behaviour/cognition

European bison, insight, Köhler, problem solving, ungulates, *Bison bonasus*

**Author for correspondence:**
Alvaro L. Caicoya
e-mail: alocaico@gmail.com

†These authors equally contributed to the paper.

# Problem solving in European bison (*Bison bonasus*): two experimental approaches

Alvaro L. Caicoya[1,2], Montserrat Colell[1,2,†],
Conrad Ensenyat[3] and Federica Amici[4,5,†]

[1]Department of Clinical Psychology and Psychobiology, Faculty of Psychology, and [2]Institute of Neurosciences, University of Barcelona, Barcelona, Spain
[3]Barcelona Zoo, Barcelona, Spain
[4]Junior Research Group 'Primate Kin Selection', Institute of Biology, Faculty of Bioscience, Pharmacy and Psychology, University of Leipzig, Leipzig, Germany
[5]Department of Human Behavior, Ecology and Culture, Max Planck Institute for Evolutionary Anthropology, Leipzig, Germany

ALC, 0000-0001-8836-3313

The ability to solve novel problems is crucial for individual fitness. However, studies on problem solving are usually done on few taxa, with species with low encephalization quotient being rarely tested. Here, we aimed to study problem solving in a non-domesticated ungulate species, European bison, with two experimental tasks. In the first task, five individuals were presented with a hanging barrel filled with food, which could either be directly accessed (control condition) or which could only be reached by pushing a tree stump in the enclosure below it and stepping on it (experimental condition). In the second task, five individuals were repeatedly fed by an experimenter using a novel bucket to retrieve food from a bag. Then, three identical buckets were placed in the enclosure, while the experimenter waited outside with the bag without feeding the bison, either with a bucket (control condition) or without it (experimental condition). In the first task, no bison moved the stump behind the barrel and/or stepped on it to reach the food. In the second task, two individuals solved the task by pushing the bucket within the experimenter's reach, twice in the experimental and twice in the control condition. We suggest that bison showed a limited ability to solve novel problems, and discuss the implications for their understanding of the functional aspects of the tasks.

# 1. Background

The ability to spontaneously solve novel problems plays a crucial role in the survival and fitness of individuals, by facilitating the

exploration of new resources, the use of novel strategies and the invasion of new niches [1,2]. Such an ability might be especially relevant in complex dynamic environments, or when socio-ecological conditions rapidly change, as the result of biotic and abiotic factors, and anthropogenic disturbance [3–5].

The study of problem solving has fascinated scientists for at least a century [6]. However, experimental studies on problem solving have so far only included few taxa. To date, most studies have been conducted on primates [7–11], birds [12–15], canines [16–18] and rodents [19]. Some notable exceptions include elephants [20], dolphins [21] and different carnivore species [22–25].

In recent years, however, some researchers have also started investigating problem solving in taxa with low encephalization quotient (EQ), such as ungulates [26–30]. These studies have shown that ungulates can solve novel problems (e.g. detour tasks) and even learn from humans how to solve them [28]. These studies are especially interesting, because they confirm that, even in mammals, the ability to solve novel problems is not limited to species with large brains [31,32]. However, it is possible that other species might perform differently. To date, for instance, problem-solving tasks have only been conducted on domesticated ungulate species. Through domestication, however, the ability to solve novel problems might have dramatically changed. On the one hand, as an adaptive consequence of human selection, domestic species might show a preference for novelty [33,34], which is linked to higher exploration and ability to innovate [7,35–38]. On the other hand, domestication has led to a reduction in brain size (which, in ungulates, ranges from 14 to 24% [39]). Therefore, the inclusion of non-domesticated species is especially important to better understand the socio-ecological conditions that might favour the emergence of problem-solving skills.

To fill this gap, we tested problem solving in a non-domesticated species, European bison (*Bison bonasus*). Nowadays, bison live in Eastern European forests [40], although historically they were distributed throughout Europe [41]. Despite many physiological adaptations to grazing, like their teeth microwear [42], European bison are mainly browsers, and this ability to adapt to a novel diet might have been crucial for their survival, as they are now the largest living herbivores in Europe [43]. This species is an ideal candidate to study problem solving, because bison live in relatively complex social systems: herds include up to 30 individuals and are characterized by high levels of fission–fusion [44], which has been proposed as a major driver of enhanced cognition [45]. Bison often migrate and form hierarchical groups which are maintained over many years, whose social structure flexibly changes across seasons [46]. Moreover, bison might engage in solitary and social play relatively often (as we observed by comparing more than 20 ungulate species during a set of studies; A. L. Caicoya, personal observation, 2019–2020), a behaviour that has been linked to behavioural innovation [47]. The genus bison has an EQ of 0.85, which is slightly higher than their close relatives like *Bos grunniensis* (EQ = 0.76), *B. javanicus* (EQ = 0.78), *B. taurus* (EQ = 0.55) or even *Equus caballus* (EQ = 0.78) [48,49], but clearly lower than other species usually tested in problem-solving studies, such us chimpanzees (*Pan troglodytes*, EQ = 2.48) or *Homo sapiens* (EQ = 6.62) [49].

In this study, we aimed to assess if a non-domesticated ungulate species with relatively low EQ [50,51], would spontaneously innovate in an experimental context. For this reason, we presented a group of captive bison with two different problem-solving tasks. In the first task, we adapted a classic experimental protocol [6] to ensure ecological relevance. Bison were presented with a familiar plastic barrel hanging from a branch: the barrel was filled with carrots, from which they regularly fed by shaking it with the muzzle or horns, letting carrots fall out of its holes and on the ground. In the experimental condition, however, the barrel was too high to be reached, but bison could access it by moving a tree stump in the enclosure and climbing on it to shake the barrel. As bison have adapted to browse and can feed on food hanging from relatively high branches, this task should be ecologically relevant for them. When tested with a similar set-up, one elephant [20] and several chimpanzees could successfully solve the task [6], while seven sloth bears failed despite being provided with several social and non-social cues [25]. In the second task, bison went through a training phase in which the experimenter repeatedly fed them with small pieces of dry carob, using a novel bucket as a scoop to retrieve the carob from a bag. In the experimental condition, the experimenter approached the bison with the bag, but the bucket had been previously placed in the enclosure, so that bison had to push it within the experimenter's reach to be fed. We hypothesized that, if bison understood the relevant aspects of the tasks (i.e. return a bucket to the experimenter to feed on carob and get on a tree stump in order to reach a barrel filled with carrots), they should have more frequently interacted with the tree stump/bucket and moved them towards the barrel/fence when this was needed to access the food (as compared to control conditions in which the objects were not functional).

# 2. Material and methods

## 2.1. Subjects

We tested five European bison (*B. bonasus*) housed at the zoo of Barcelona, ranging from 6 to 30 years of age. All study individuals were habitually fed on a diet of grasses. Individuals had little experience with experimental procedures, having only been tested in an object permanence task. Crucially, none of the individuals had ever been trained by the experimenters or by the zookeepers to return objects in the enclosure. The tasks were carried out in the external facilities of the bison, and their usual management was not changed due to our tasks. The bison enclosure size was $617 \, m^2$ and did not include many visual barriers (see figures 1, 2 and 3 for photos of the enclosure). The bison were not separated during the tasks, as testing took place while all bison could freely move inside their enclosure.

## 2.2. Task 1. Setup and procedure

In Task 1, we used a plastic barrel of approximately 100 cm height and 60 cm diameter, and a tree stump of 34 cm height, 46 cm diameter and 14 kg. Bison had extensive experience with the barrel, since it was a common enrichment in their facilities. The barrel habitually hung from a branch, and individuals shook it with their muzzle/horns to let carrots fall out of its holes and on the ground, where they were eaten. By contrast, bison had no experience with the stump used for the experiment, although their facilities contained bigger tree trunks, which bison easily moved around the enclosure. Every morning, when bison were moved in the internal facilities in order to clean the external ones, the experimenter hung the stump 5 m from the barrel and filled the barrel with carrots. In the experimental condition, the barrel hung approximately 20 cm higher than usually, so that bison could only reach it by moving the stump below it and standing on it (figure 1). Crucially, before this study was conducted, all bison had been repeatedly observed putting their front hooves on the feeders in their facilities and stand on them for apparently no reason, suggesting that standing on objects belonged to their natural repertoire (figure 2). In the control condition, the barrel was hanging as usual, so that the stump, despite being also present in the enclosure, was not functional to retrieve the carrots (figure 3). If bison did not solve the experimental condition after six sessions, the same condition was repeated, but placing the stump under the barrel, so that individuals only had to get on the stump to hit the barrel and retrieve the food. Each session lasted for 24 h or until the barrel ran out of carrots. We administered a total of 20 experimental and 13 control sessions in a pseudo-randomized order (i.e. ensuring that no more than two identical conditions were administered in a row). The number of sessions differed between the two conditions to meet the management needs of the zoo.

## 2.3. Task 2. Setup and procedure

In Task 2, all bison first went through a training phase of 10 days, in which the experimenter repeatedly used a novel $35 \times 28 \times 18$ cm blue bucket to feed individuals with small pieces of dry carob—a highly preferred food item (figure 4). The bucket was ostensibly used as a scoop, with the experimenter retrieving the carob from a filled bag directly with the bucket, and never doing it in any other way, so that the bison could associate that the bucket was essential to be fed with the carob. In the experimental condition, the experimenter placed three identical buckets (to avoid monopolization by single individuals) in the internal facilities, when the bison were out of view in the external facilities. The buckets were placed 3 m from the fence and 3 m from each other. When the bison entered the internal enclosure, the experimenter approached the fence, stopped 2 m from it, placed the bag with carob on the floor and then strolled around as though looking for something (i.e. the bucket) for 5 min. After that, the experimenter left the area for 5 further minutes (leaving the bag filled with carob visible on the ground), and then came back and repeated the procedure for 5 more minutes. If one bison pushed the bucket within the experimenter's reach during these 15 min, the experimenter immediately reached for it through the fence and gave the carob to the bison (figure 5). If the bucket was not returned, the experimenter took the bag with the carob and left. Buckets were then collected from the enclosure by the keepers, after the experimenter had left. In the control condition, we repeated the same procedure as in the experimental one, with the only exception being that the experimenter also had a bucket. We administered a total of three experimental and four control sessions, starting with the control condition and alternating them. In this way, we could compare performance in experimental sessions to performance in control sessions which were administered both before and after the experimental ones. Please refer to electronic supplementary material S1 for a video of a bison returning the bucket in an experimental session.

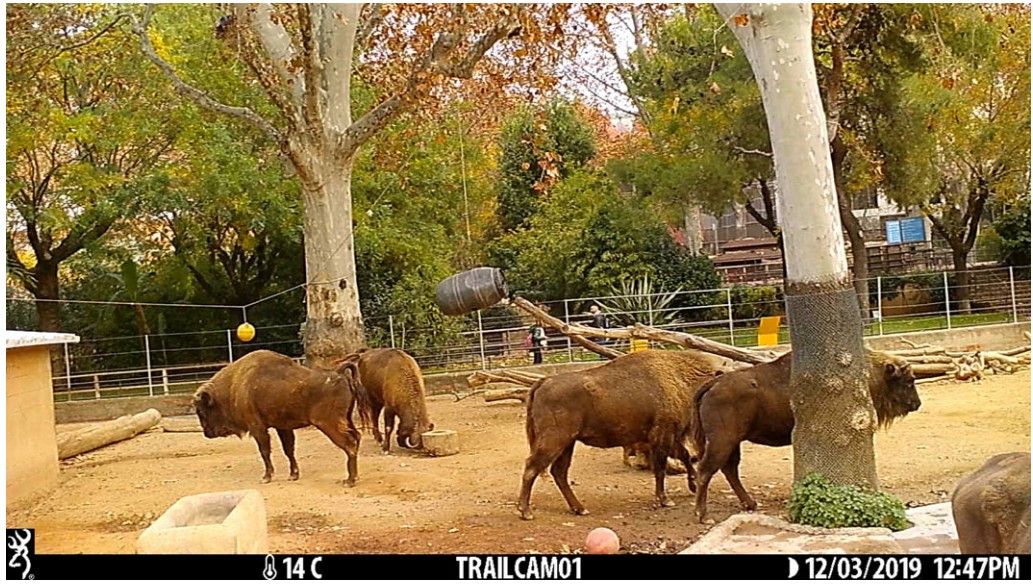

**Figure 1.** Experimental setup for Task 1. The second bison from the left is moving the stump, while the barrel is low enough to be reached without the stump (i.e. control condition).

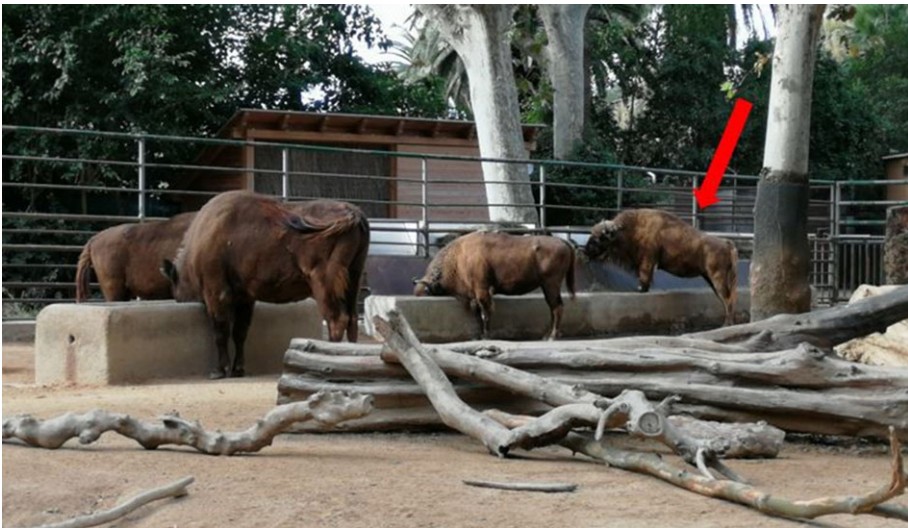

**Figure 2.** A bison with its front hooves on the feeder. Bison habitually stood on objects.

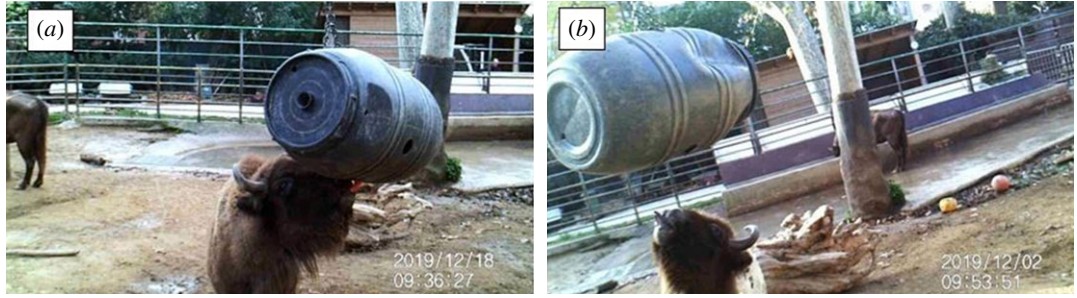

**Figure 3.** Setup of the (*a*) control and (*b*) experimental conditions. The barrel hangs within and without reach, respectively.

## 2.4. Coding

All sessions were video-recorded. In Task 1, we used two camera traps, one filming the whole area and one focusing on the barrel from a 2 m-distance. In Task 2, we used a video camera standing on a tripod

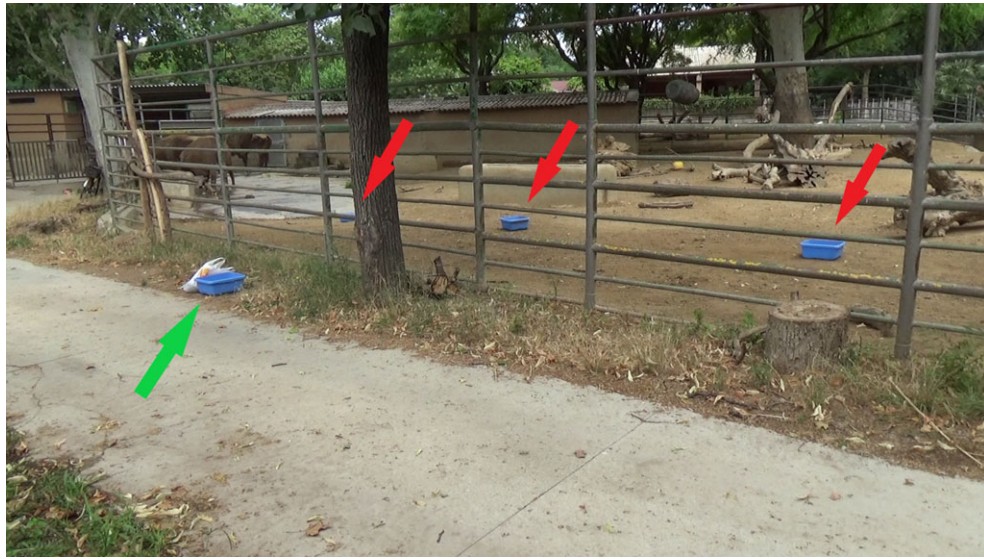

**Figure 4.** Experimental setup for Task 2 in a control session. There are three buckets inside the facilities (red arrows), at 3 m from the fence. The experimenter had another bucket outside the bison enclosure (green arrow).

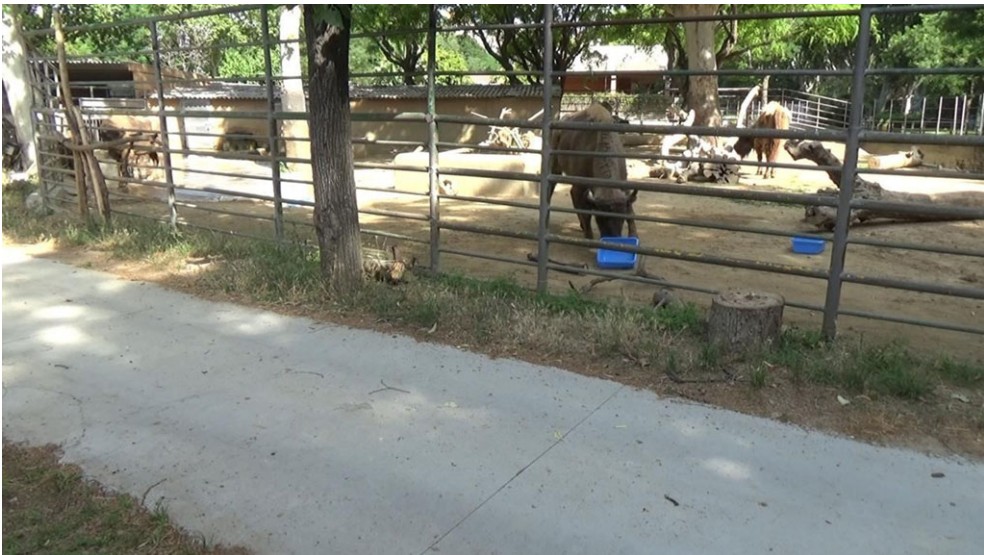

**Figure 5.** A bison returning a bucket in the experimental session, when the experimenter left the area. Please refer to electronic supplementary material S1 for the whole video of the bison returning the bucket.

3 m behind the experimenter's back, filming the fence with the three buckets on the bison's side. In Task 1, we watched the videos to code: (i) whether individuals solved the task (i.e. climbed on the stump and got the carrots); (ii) the latency to first approach (i.e. stand below) the barrel (as a measure of motivation); (iii) the latency to first interact with the stump (i.e. touching, moving or sniffing it); (iv) the exact duration of time spent moving the stump or (v) otherwise interacting with it. In Task 2, we watched the videos to code: (i) whether individuals solved the task (i.e. pushed the bucket within the experimenter's reach and got the carob); (ii) latency to first interact with the bucket (i.e. touching, moving or sniffing it); (iii) the exact duration of time spent interacting with the bucket, (iv) moving it in the correct direction (i.e. toward the fence) or (v) in any other direction.

## 2.5. Statistical analyses

Analyses were conducted using generalized linear mixed models [52] with the glmmTMB package (v. 1.0.1 [53]) in R (R Core Team, v. 3.5.0). In all models, we entered one line per individual and session. Models 1–4 assessed performance in Task 1. In particular, we assessed whether the condition (i.e. Experimental or

Control) predicted the latency to first approach the barrel (Model 1), the latency to first interact with the stump (Model 2), whether they moved the stump (Model 3) or otherwise interacted with it (Model 4). In all models, we controlled for the distance between stump and barrel (i.e. 5 m or 0 m) and session number, we included individual identity as random factor and (in Models 3–4) session duration as offset term. Models 5–7 assessed performance in Task 2. In particular, we assessed whether condition (i.e. Experimental or Control) predicted the latency to first interact with the bucket (Model 5), the probability of moving the bucket (Model 6) or otherwise interacting with it (Model 7). In all models, we included individual identity as random factor and session duration as offset term. Finally, as there were no instances of success in Task 1 and only a few in Task 2, no statistical analyses were run for this variable.

All models were run with a binomial structure, except for models 1, 2 and 5, which had a Gaussian distribution. We used likelihood ratio tests [54] to compare full models containing all predictors with null models containing only control predictors, offset terms and random factors. When full models significantly differed from null models, likelihood ratio tests were conducted to obtain the $p$-values for each test predictor via single-term deletion using the R function drop1 [55]. We detected no convergence issues. To rule out collinearity, we determined the variance inflation factor (VIF) [56], which was minimal (maximum VIF across all models = 1.35).

# 3. Results

None of the individuals solved Task 1. In Task 2, however, individuals pushed the bucket within the experimenter's reach in 4 of the 30 administered sessions: one individual (Elipse) did it twice, in two Experimental sessions, while another individual (Verde) did it twice, in two Control sessions.

Models 1–4 assessed performance in Task 1. For Model 1, the full-null model comparison was significant (GLMM: $\chi^2 = 22.48$, d.f. = 1, $p < 0.001$), with latency to approach the barrel being significantly higher in the Experimental than in the Control condition (table 1). By contrast, there was no significant difference between full and null model for Model 2 (GLMM: $\chi^2 = 0.08$, d.f. = 1, $p = 0.778$), Model 3 (GLMM: $\chi^2 = 0.62$, d.f. = 1, $p = 0.430$) and Model 4 (GLMM: $\chi^2 = 0.02$, d.f. = 1, $p = 0.886$). In particular, condition neither predicted the latency to first interact with the stump (Model 2), nor the probability of moving it (Model 3) or otherwise interacting with it (Model 4; table 1).

Models 5–7 assessed performance in Task 2. There was no significant difference between full and null model for Model 5 (GLMM: $\chi^2 = 0.28$, d.f. = 1, $p = 0.598$), Model 6 (GLMM: $\chi^2 = 1.42$, d.f. = 1, $p = 0.233$) and Model 7 (GLMM: $\chi^2 = 0.01$, d.f. = 1, $p = 0.978$). In particular, condition failed to predict the latency to first interact with the bucket (Model 5), and the probability of moving the bucket (Model 6) or otherwise interacting with it (Model 7; see table 1).

# 4. Discussion

In our study, bison showed a limited ability to solve novel problems. In the first task, none of the five study individuals moved the tree stump behind the barrel with food and/or stepped on it to reach for the food when it was out of reach (experimental condition). Bison approached the barrel significantly later when the barrel was out of reach, as compared to when it was accessible (control condition). However, the latency to first interact with the stump and the probability of moving it or otherwise interacting with it did not significantly change across conditions. In the second task, two individuals solved the task by pushing the bucket within the experimenter's reach: one individual did it twice, when the experimenter had no bucket (experimental condition), and one did it twice, when the experimenter had the bucket but still refrained from feeding the bison (control condition). As in Task 1, the latency to first interact with the bucket and the probability of moving it or otherwise interacting with it did not change across conditions. Please refer to electronic supplementary material S1 for a video of a bison returning the bucket in an experimental session.

In the first task, none of the study individuals mastered the task. Individuals interacted in a similar way with the stump in the experimental and control condition, suggesting that they did not understand the functional value of the stump to solve the task and access the food in the experimental condition. Bison only differed between conditions in their latency to approach the barrel, which was higher in the experimental condition. This suggests that bison were less motivated to approach the barrel when it was out of reach, and this could be interpreted in at least three different ways. First, it is possible that the motivation to approach the barrel in the experimental condition decreased through time, as bison could not retrieve food and therefore learned that food was not accessible when the barrel was hanging higher

**Table 1.** Summary of results. Results of the models run, including estimates, standard errors (s.e.), confidence intervals (CIs) and *p*-values for each test and control predictor (in parentheses, the reference category). Significant test predictors are in bold, control predictors in italics. All models included individual identity as random effect, and Models 3, 4, 6 and 7 also included session duration as offset. The asterisks denote significant *p*-values for the test predictors. All models had a binomial distribution, except for Models 1, 2 and 5, which had a Gaussian distribution.

| model | estimate | s.e. | 2.5% CI | 97.5% CI | *p*-value |
|---|---|---|---|---|---|
| M1: Latency to first approach the barrel (Task 1) | | | | | |
| intercept | 714.80 | 181.30 | 359.46 | 1070.14 | — |
| **condition** | 439.90 | 89.54 | 264.40 | 615.40 | <0.001* |
| *distance* | 9.69 | 96.47 | −179.40 | 198.77 | 0.920 |
| *session number* | −19.17 | 13.51 | −45.65 | 7.30 | 0.157 |
| M2: Latency to first interact with the tree stump (Task 1) | | | | | |
| intercept | 1175.03 | 83.28 | 1011.81 | 1338.25 | — |
| condition | −21.96 | 77.86 | −174.57 | 130.65 | 0.778 |
| *distance* | −207.80 | 83.89 | −372.22 | −43.38 | 0.013 |
| *session number* | 38.67 | 11.75 | 15.64 | 61.69 | <0.001 |
| M3: Probability of moving the tree stump (Task 1) | | | | | |
| intercept | −13.43 | 0.89 | −15.18 | −11.68 | — |
| condition | −0.73 | 0.94 | −2.58 | 1.11 | 0.436 |
| *distance* | −0.94 | 0.97 | −2.84 | 0.96 | 0.335 |
| *session number* | −0.11 | 0.18 | −0.47 | 0.25 | 0.547 |
| M4: Probability of otherwise interacting with the tree stump (Task 1) | | | | | |
| intercept | −12.15 | 0.47 | −1.31 | −1.12 | — |
| condition | 0.06 | 0.43 | −7.75 | 8.97 | 0.887 |
| *distance* | 0.41 | 0.46 | −4.85 | 1.3 | 0.371 |
| *session number* | −0.21 | 0.08 | −3.62 | −6.15 | 0.006 |
| M5: Latency to first interact with the bucket (Task 2) | | | | | |
| intercept | 14 | 1.96 | 10.15 | 17.85 | — |
| condition | 0.74 | 1.39 | −1.98 | 3.46 | 0.597 |
| *session number* | −1.68 | 0.85 | −3.34 | −0.01 | 0.049 |
| M6: Probability of moving the bucket (Task 2) | | | | | |
| intercept | −8.94 | 2.42 | −13.69 | −4.2 | — |
| condition | −1.46 | 1.31 | −4.04 | 1.1 | 0.264 |
| M7: Probability of otherwise interacting with the bucket (Task 2) | | | | | |
| intercept | −8.76 | 1.17 | 11.07 | −6.46 | — |
| condition | −0.03 | 1.13 | −2.25 | 2.19 | 0.978 |

on the tree. However, this explanation is unlikely, in that the latency to approach the barrel did not vary through time (i.e. session had no significant effect on the latency to approach the barrel in Model 1; see table 1). Second, it is possible that the motivation to approach the barrel was lower in the experimental condition from the very beginning, because bison understood that food could not be reached when the barrel was hanging higher, despite failing to understand how to reach for it. Third, it is possible that bison simply required longer to approach the barrel when it was hanging higher, because they were not used to see the barrel in that position and were thus simply reacting fearfully to the novel situation [57,58]. In the future, further studies should better assess the role played by neophobia in the bison's behaviour, to contrast these different hypotheses.

In the second task, two individuals spontaneously pushed the bucket within the experimenter's reach. Crucially, none of the study individuals had been previously trained to return objects in the enclosure.

Moreover, the zookeepers in Barcelona reported having never asked bison to give back objects left in their enclosures, nor having rewarded them in any way for pushing objects in the enclosure, and were indeed highly surprised by the behaviour of the animals. While one individual pushed the bucket within reach twice in the experimental condition (i.e. when the experimenter had the bag with carob, but no bucket, so that the bucket was functional to solve the task), another individual did it in the control condition (i.e. when the experimenter had the bag and the bucket, but still refrained from feeding the bison). These results may be explained in two different ways. First, it is possible that bison failed to understand the contingencies of the task, and then simply pushed the bucket toward the observer, without understanding its function. However, the bison did not simply interact with the bucket, but directly pushed it toward the experimenter for 3 m, until the experimenter could reach for it through the fence. Therefore, this suggests that the bison understood that the bucket was required to obtain the food. Second, it is possible that the bison, despite understanding the functional value of the bucket, failed to differentiate between experimental and control condition. In particular, bison could have simply reacted to the experimenter providing no food by pushing the functional object to food retrieval, regardless of whether the experimenter had it already. Indeed, our study subjects had 'nothing to lose' by trying to push the bucket toward the experimenter. If this is true, our results show that bison can successfully solve novel tasks by showing a general basic understanding of object functionalities. Future studies should ideally include more control conditions to better disentangle which task contingencies are taken into account by bison when solving novel problems, by for instance including control sessions in which novel but non-functional objects are placed in the enclosure. Finally, it is also possible that the bison pushing the bucket twice in the control session did it after socially learning to do it (i.e. after observing the other individual pushing the bucket in the experimental sessions). However, this explanation is unlikely, because the bucket was first returned in a control session (the first one), then in an experimental one (the fourth), then again in a control session (the fifth) and finally in an experimental one (the sixth). Please refer to electronic supplementary material S1 for a video of a bison returning the bucket in an experimental session.

Our study should clearly be considered as a first preliminary step in the investigation of problem solving in non-domesticated ungulate species. Overall, it confirms ungulates as a promising model to study innovation and, more generally, cognition [28,30,59–62]. Despite their relatively small brain size [50,51], bison showed some ability to solve novel problems, although their exact understanding of the functional aspects of the tasks is unclear. Whether specific socio-ecological characteristics are linked to the emergence of enhanced cognitive abilities, and/or to a more general attitude toward novelty, is a question that remains to be addressed.

Ethics. The Barcelona Zoo controlled and approved all the procedures. Given that bison participated on a completely voluntary basis, and no invasive procedures were used, no formal approval was required. During the task, individuals were never food or water deprived, and motivation to participate was ensured exclusively by the use of highly preferred food (i.e. carrots and carob). The experiments thus provided a form of enrichment for the individuals and did not present any risks or adverse effects.

Data accessibility. The data are provided in electronic supplementary material [63].

Authors' contributions. A.L.C. carried out the field work with the help of bachelor students, conceived the study, participated in data analysis, participated in the design of the study and drafted the manuscript; M.C. participated in designing the study, coordinated the study and helped draft the manuscript. C.E. participated in the design and coordination of the study. F.A. designed the study, carried out the statistical analyses and helped draft the manuscript. All authors gave final approval for publication and agree to be held accountable for the work performed therein.

Competing interests. We declare we have no competing interests.

Funding. This research was funded by a PRIC grant, 2019/2020, from the Fundación Zoo de Barcelona.

Acknowledgements. We thank Carme Mora Rueda and Marta Portolà Lodoso for data collection in the first task, and Marta Portolá for the selection of the images used for this article. We thank the staff at the bison facilities in Barcelona, and in particular Pilar Padilla, Daniel Muñoz, Núria Moreno, Luis Mesa, José Ramón Palomar, Sergi Baselga, Nerea Lebrero, and all the others for endless support and cooperation.

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
