## [Peer Review File · Royal Society Open Science]

Review History

RSOS-201901.R0 (Original submission)

Review form: Reviewer 1

Is the manuscript scientifically sound in its present form?

Yes

Are the interpretations and conclusions justified by the results?

Yes

Is the language acceptable?

Yes

Do you have any ethical concerns with this paper?

No

Have you any concerns about statistical analyses in this paper?

No

Recommendation?

Accept with minor revision (please list in comments)

Comments to the Author(s)

This is an excellent study of cognition in a species that is understudied. The procedures and methods are sound. I have some suggestions for clarification which I have added below.

Line 24. The plural of bison is bison. Throughout the paper.

Line 43. Cut "for instance"

Line 88. Should be "ecological relevance"

Line 121. Please add more details about the enclosure: size, visibility of herd members, etc.

Line 124. In Procedure, need to state that the bison were tested with others in the enclosure. It is obvious from the photo but should be stated as part of procedures.

Line 231. Should be "Variance Inflation Factor (VIF)" and "maximum VIF".

Line 306. There is a third possibility: The bison that pushed the bucket in the control condition had seen the other bison move the bucket to get food. Displaying social learning, the 2nd bison pushed the bucket twice, but not receiving reinforcement, stopped pushing the bucket afterwards. Would need to know if the bison in the experimental condition completed the task first and if the second bison was in a position to observe the first. If this was not possible, it needs to be explained.

Data sheets could use more explanation of variables.

Review form: Reviewer 2

Is the manuscript scientifically sound in its present form?

Yes

Are the interpretations and conclusions justified by the results?

Yes

Is the language acceptable?

Yes

Do you have any ethical concerns with this paper?

No

Have you any concerns about statistical analyses in this paper?

No

Recommendation?

Accept with minor revision (please list in comments)

Comments to the Author(s)

This paper shows that bison fail to move a stump underneath a bucket in order to bring to within reach, but push a container within reach of an experimenter, though they do not appear to discriminate between an experimenter already with a container and one without. As the authors state this is an intriguing preliminary evidence for bison being capable of problem solving, but controls need to be run to understand what bison understand about the problems presented to them.

Line 63 It is not clear what these percentages refer to, or why the authors use the word 'might' in this sentence? Is there clear evidence that domestication reduces absolute brain size?

Line 77 Given this claim is based only on personal observation a bit more caution in regard to this claim is merited.

Line 101 What were the relevant aspects of the task exactly? Some precision here would be helpful.

Line 134 'hung'

Line 144 Why was the number of experimental and control sessions not balanced?

Line 165 Why 3 buckets?

Line 177 it is not clear how this is an appropriate control, the experimenter has a bucket but does not use it, would a rational human not assume that giving then another bucket might be helpful in gaining the food?

Line 178 Why was the number of experimental and control sessions not balanced?

Line 248 Please make clear this analysis pertains to Study 2.

Line 283 please reword "it might be"

Line 317 What control conditions might be appropriate to run

Decision letter (RSOS-201901.R0)

Dear Mr Caicoya

On behalf of the Editors, we are pleased to inform you that your Manuscript RSOS-201901 "Problem solving in European bison (*Bison bonasus*): two experimental approaches" has been accepted for publication in Royal Society Open Science subject to minor revision in accordance with the referees' reports. Please find the referees' comments along with any feedback from the Editors below my signature.

Please submit your revised manuscript and required files (see below) no later than 7 days from today's (ie 26-Mar-2021) date. Note: the ScholarOne system will 'lock' if submission of the revision is attempted 7 or more days after the deadline. If you do not think you will be able to meet this deadline please contact the editorial office immediately.

Please note article processing charges apply to papers accepted for publication in Royal Society Open Science (<https://royalsocietypublishing.org/rsos/charges>). Charges will also apply to papers transferred to the journal from other Royal Society Publishing journals, as well as papers

submitted as part of our collaboration with the Royal Society of Chemistry (<https://royalsocietypublishing.org/rsos/chemistry>). Fee waivers are available but must be requested when you submit your revision (<https://royalsocietypublishing.org/rsos/waivers>).

on behalf of Professor Marcelo Sanchez (Associate Editor) and Kevin Padian (Subject Editor)
openscience@royalsociety.org

Editor Comments to Author:

Thanks for your submission. As you see, the reviewers are generally happy with the paper but they have some minor comments that we ask you to incorporate into your final version. Please indicate in your revision that you have done this, so that we can proceed with publication without delay.

Reviewer comments to Author:

Reviewer: 1
Comments to the Author(s)

This is an excellent study of cognition in a species that is understudied. The procedures and methods are sound. I have some suggestions for clarification which I have added below.

Line 24. The plural of bison is bison. Throughout the paper.

Line 43. Cut "for instance"

Line 88. Should be "ecological relevance"

Line 121. Please add more details about the enclosure: size, visibility of herd members, etc.

Line 124. In Procedure, need to state that the bison were tested with others in the enclosure. It is obvious from the photo but should be stated as part of procedures.

Line 231. Should be "Variance Inflation Factor (VIF)" and "maximum VIF".

Line 306. There is a third possibility: The bison that pushed the bucket in the control condition had seen the other bison move the bucket to get food. Displaying social learning, the 2nd bison pushed the bucket twice, but not receiving reinforcement, stopped pushing the bucket afterwards. Would need to know if the bison in the experimental condition completed the task first and if the second bison was in a position to observe the first. If this was not possible, it needs to be explained.

Data sheets could use more explanation of variables.

Reviewer: 2
Comments to the Author(s)

This paper shows that bison fail to move a stump underneath a bucket in order to bring to within reach, but push a container within reach of an experimenter, though they do not appear to discriminate between an experimenter already with a container and one without. As the authors state this is an intriguing preliminary evidence for bison being capable of problem solving, but controls need to be run to understand what bison understand about the problems presented to them.

Line 63 It is not clear what these percentages refer to, or why the authors use the word 'might' in this sentence? Is there clear evidence that domestication reduces absolute brain size?

Line 77 Given this claim is based only on personal observation a bit more caution in regard to this claim is merited.

Line 101 What were the relevant aspects of the task exactly? Some precision here would be helpful.

Line 134 'hung'

Line 144 Why was the number of experimental and control sessions not balanced?

Line 165 Why 3 buckets?

Line 177 it is not clear how this is an appropriate control, the experimenter has a bucket but does not use it, would a rational human not assume that giving then another bucket might be helpful in gaining the food?

Line 178 Why was the number of experimental and control sessions not balanced?

Line 248 Please make clear this analysis pertains to Study 2.

Line 283 please reword "it might be"

Line 317 What control conditions might be appropriate to run

===PREPARING YOUR MANUSCRIPT===

If you have been asked to revise the written English in your submission as a condition of publication, you must do so, and you are expected to provide evidence that you have received language editing support. The journal would prefer that you use a professional language editing service and provide a certificate of editing, but a signed letter from a colleague who is a native speaker of English is acceptable. Note the journal has arranged a number of discounts for authors

using professional language editing services
(<https://royalsociety.org/journals/authors/benefits/language-editing/>).

===PREPARING YOUR REVISION IN SCHOLARONE===

-- If you have uploaded ESM files, please ensure you follow the guidance at <https://royalsociety.org/journals/authors/author-guidelines/#supplementary-material> to include a suitable title and informative caption. An example of appropriate titling and captioning may be found at https://figshare.com/articles/Table_S2_from_Is_there_a_trade-

off_between_peak_performance_and_performance_breadth_across_temperatures_for_aerobic_sc
ope_in_teleost_fishes_/3843624.

Author's Response to Decision Letter for (RSOS-201901.R0)

See Appendix A.

Decision letter (RSOS-201901.R1)

Dear Mr Caicoya,

It is a pleasure to accept your manuscript entitled "Problem solving in European bison (*Bison bonasus*): two experimental approaches" in its current form for publication in Royal Society Open Science.

on behalf of Professor Marcelo Sanchez (Associate Editor) and Kevin Padian (Subject Editor)
openscience@royalsociety.org

Editor comments:

The AE is pleased with your revisions and so we are happy to accept your manuscript. Best wishes.

Appendix A

Reviewer: 1

Comments to the Author(s)

This is an excellent study of cognition in a species that is understudied. The procedures and methods are sound. I have some suggestions for clarification which I have added below.

- Thank you very much for your appreciations, we hope the new changes clarify all the pending details.

Line 24. The plural of bison is bison. Throughout the paper.

- We now only use “bison” throughout the manuscript.

Line 43. Cut “for instance”

- Done. Line 43 (line numbers refer to the MS without track changes)

Line 88. Should be “ecological relevance”

- Changed. Line 88

Line 121. Please add more details about the enclosure: size, visibility of herd members, etc.

- We have added this information: “The bison enclosure size was 617m² and did not include many visual barriers (see Fig. 1, 2 and 3 for some photos of the enclosure).” Line 123

Line 124. In Procedure, need to state that the bison were tested with others in the enclosure. It is obvious from the photo but should be stated as part of procedures.

- We have added this information: “The bison were not separated during the tasks, as testing took place while all bison could freely move inside their enclosure.” Line 124

Line 231. Should be “Variance Inflation Factor (VIF)” and “maximum VIF”.

- We have changed this. Line 241

Line 306. There is a third possibility: The bison that pushed the bucket in the control condition had seen the other bison move the bucket to get food. Displaying social learning, the 2nd bison pushed the bucket twice, but not receiving reinforcement, stopped pushing the bucket afterwards. Would need to know if the bison in the experimental condition completed the task first and if the second bison was in a position to observe the first. If this was not possible, it needs to be explained.

- The bucket was returned four times: in the first session (control), in the fourth session (experimental), in the fifth session (control) and finally in the sixth trial (experimental one). Because of this order, we think it is unlikely that bison pushed the bucket in the control sessions after socially learning it from the others in experimental sessions. However, this is an interesting hypothesis, and we have now discuss it in the MS, thanks. Line 330

Data sheets could use more explanation of variables.

- We now provide all the data in an excel file with more explanations. Thanks for pointing this out.

Reviewer: 2

Comments to the Author(s)

This paper shows that bison fail to move a stump underneath a bucket in order to bring to within reach, but push a container within reach of an experimenter, though they do not appear to discriminate between an experimenter already with a container and one without. As the authors state this is an intriguing preliminary evidence for bison being capable of problem solving, but controls need to be run to understand what bison understand about the problems presented to them.

- Thank you very much for your kind words

Line 63 It is not clear what these percentages refer to, or why the authors use the word ‘might’ in this sentence? Is there clear evidence that domestication reduces absolute brain size?

- We have deleted the word “might”. We think there is abundant literature on the link between brain size reduction and domestication, as stated for example in this extract from “Pathways to Animal Domestication” (38): *Numerous studies have noted a systematic reduction in the overall size of brains in domestic animals compared with their wild progenitors, (Kruska 1988, 1996, Plogmann and Kruska 1990, Ebinger 1995, Ebinger and Röhrs 1995). Within broad classes of domesticated mammals, there is a positive correlation between the degree of encephalization (brain mass above that related to an animal’s body mass) and brain size reduction. Mammals with larger brains seem to have experienced the greatest degree of brain size reduction, whereas smaller-brained mammals may experience little or no overall reduction in brain size with domestication (Kruska 1988).*

Line 77 Given this claim is based only on personal observation a bit more caution in regard to this claim is merited.

- We have rephrased it, thanks. Line 76 (line numbers refer to the MS without track changes)

Line 101 What were the relevant aspects of the task exactly? Some precision here would be helpful.

- We have added a line here to be more precise: “(i.e. return a bucket to the experimenter to feed on carob and get on a tree stump in order to reach a barrel filled with carrots)”. Line 102

Line 134 ‘hung’

- Changed. Line 135

Line 144 Why was the number of experimental and control sessions not balanced?

- The barrel was routinely used by the keepers in Barcelona as a sort of enrichment for the bison, so we had to adapt to the daily management routines of the zoo to conduct our task (e.g. conducting experimental sessions when zoo keepers deemed important that bison didn't retrieve much food). We have added some of this information in the text. Line 148

Line 165 Why 3 buckets?

- By providing more than one bucket, we aimed to ensure that all individuals could interact with them, without the more dominant individuals monopolizing access to them. Line 175

Line 177 it is not clear how this is an appropriate control, the experimenter has a bucket but does not use it, would a rational human not assume that giving them another bucket might be helpful in gaining the food?

- Our rationale was that, if the experimenter has the bucket but does not feed the bison, the bison should infer that there is "something wrong other than the bucket". However, it is true that the bison had "nothing to lose" by nonetheless pushing the bucket to the experimenter. We now briefly discuss this in the MS. Thanks for pointing this out. Line 324

Line 178 Why was the number of experimental and control sessions not balanced?

- We started with a control condition to have a sort of baseline about animals' behaviour before being exposed to the experimental set-up. However, we also wanted to have a final control session, to ensure comparisons between experimental and control conditions also when control sessions followed experimental sessions. Luckily, the model controls for this. We now briefly explain this in the MS. Line 189

Line 248 Please make clear this analysis pertains to Study 2.

- Done, Line 257

Line 283 please reword "it might be"

- We have changed it into "could". Line 292

Line 317 What control conditions might be appropriate to run

- We have added an example "for instance including control sessions in which novel but non-functional objects are placed in the enclosure". Line 329